# Immunomodulation of Cancer Cells Using Autologous Blood Concentrates as a Patient-Specific Cell Culture System: A Comparative Study on Osteosarcoma and Fibrosarcoma Cell Lines

**DOI:** 10.3390/bioengineering11040303

**Published:** 2024-03-23

**Authors:** Eva Dohle, Kamelia Parkhoo, Francesco Bennardo, Lena Schmeinck, Robert Sader, Shahram Ghanaati

**Affiliations:** 1FORM—Frankfurt Orofacial Regenerative Medicine, Department for Oral, Cranio-Maxillofacial and Facial Plastic Surgery, Medical Center of the Johann Wolfgang Goethe University, 60590 Frankfurt, Germany; k.parkhoo@gmx.de (K.P.); lenaschmeinck@icloud.com (L.S.); sader@em.uni-frankfurt.de (R.S.); shahram.ghanaati@me.com (S.G.); 2School of Dentistry, Magna Graecia University of Catanzaro, 88100 Catanzaro, Italy; francesco.bennardo@unicz.it

**Keywords:** immunomodulation, tumor cell lines, autologous blood concentrates, proliferation, cell cycle, apoptosis

## Abstract

The understanding that tumor cells might evade immunity through various mutations and the potential of an augmented immune system to eliminate abnormal cells led to the idea of utilizing platelet-rich fibrin (PRF), a blood concentrate containing the body’s immune elements as an adjunctive therapy for localized tumors. This study is the first that evaluated the effect of PRF generated with different relative centrifugal forces (RCFs) on osteoblastic and fibroblastic tumor cell lines MG63 and HT1080 with regard to cell viability, cytokine and growth factor release, and the gene expression of factors related to the cell cycle and apoptosis. Our findings could demonstrate decreased cell proliferation of MG63 and HT1080 when treated indirectly with PRF compared to cell cultures without PRF. This effect was more distinct when the cells were treated with low-RCF PRF, where higher concentrations of growth factors and cytokines with reduced RCFs can be found. Similar patterns were observed when assessing the regulation of gene expression related to the cell cycle and apoptosis in both MG63 and HT1080 cells treated with PRF. Despite variations, there was a consistent trend of an up-regulation of tumor-suppressive genes and a down-regulation of anti-apoptotic genes in both cell types following treatment with high- and, particularly, low-RCF PRF formulations.

## 1. Introduction

In the medical and dental fields, the continuous development of regenerative concepts is essential to ensure tissue regeneration after trauma or tumor resection. In order to improve wound healing and to facilitate the process of tissue regeneration, autologous blood concentrates were introduced as a minimally invasive alternative to support tissue regeneration [1,2,3,4,5]. In this context, platelet-rich fibrin (PRF), is increasingly being used in the treatment of patients to enhance wound healing and tissue regeneration [6]. Examples of PRF’s clinical applications include bone augmentation, ridge preservation, the treatment of osteonecrosis, and soft tissue management [7,8]. In recent years, optimizing the generation of PRF has been a research focus to achieve an optimal concentration of leukocytes, platelets, and their growth factors, which are embedded in a fibrin matrix [1,9]. The introduction of the low-speed centrifugation concept (LSCC) potentially improved the regenerative capacity of PRF by reducing the relative centrifugation force (RCF) from a high towards a low range, thus increasing the content of key growth factors like transforming growth factor-ß1 (TGF-ß1), vascular endothelial growth factor (VEGF), or epidermal growth factor (EGF), as well as the number of platelets and leukocytes within the PRF [1,9,10,11]. While numerous in vitro and in vivo studies have confirmed the effectiveness of blood concentrates on healthy cells, there is still limited information about their impact on malignant cells. The strategic approach to treating cancer depends on multiple factors that should be considered, including the type of cancer, the location, the cancer stage, and the possible metastatic spread. The principal goal of tissue regeneration also applies to cancer patients, but the primary focus is typically on treating the disease itself. Even though various therapy options are available for different types of tumors, the accompanied side effects or risks often decrease the quality of life for patients and can even cause death in extreme cases. Understanding that tumor cells are cells that could escape immunity through various mutations leads to the idea that an enforced immune system might be able to help eliminate the abnormal cells in combination with other treatments like surgery or radiation.

PRF as an autologous concentrate of the blood’s immune properties, i.e., leukocytes, growth factors, cytokines, platelets, and fibrin, might therefore be an additional treatment option for localized tumors. Especially for visible carcinomas with a manageable size like in the oral cavity or the skin, PRF might be easily applied without side effects since ‘the drug’ is the body’s immune components in a concentrated form. PRF, as a booster for wound healing and regeneration, may be a promising additional treatment option for cancer patients to enhance the body’s immune response and accelerate healing mechanisms [12]. Nevertheless, the immunomodulating properties of natural biomaterials, like PRF, should be examined very critically, since they might also be ineffective regarding tumor suppression or might also favor tumor promotion.

In the context of potential tumor-suppressive impact, the effect of injectable PRF was analyzed on an in vitro monoculture of osteosarcoma cell line MG63 or fibrosarcoma cell line HT1080 [13,14] concerning cell viability, cytokine and growth factor release, and the gene expression of cell cycle- and apoptosis-regulating factors. To the best of our knowledge, this is the first study evaluating the cellular reaction of different-RCF PRF on tumor cells. The observations in this study should pave the way for future research on PRF as a patient-specific cell culture system to influence the immunomodulation of cancer cells. Therefore, further investigations into possible molecular changes and extensions to include more types of tumor cells might be useful.

## 2. Materials and Methods

### 2.1. Ethical Statement

The generation and application of the autologous blood concentrates that were used for this study were in accordance with the principle of informed consent and approved by the responsible Ethics Commission of the state of Hesse, Germany, (265/17) and all donors gave informed consent to the use of their blood for study purposes.

### 2.2. Cell Lines

For the present study, osteoblastic (MG63) and fibroblastic (HT1080) tumor cell lines were used [13,14]. After quickly thawing the cells, they were cultivated at 37 °C in a CO_2_ incubator (humidified 5% CO_2_ and 95% O_2_ atmosphere) in RPMI 1640 medium modified with the addition of 10% FBS and 1% Pen/Strep. For cell culture experimentation, approximately 2 × 10^6^ cells were seeded into one T-75 cell culture flask and cultivated for at least 24 h until use for further experimentation. After reaching confluence, cells were trypsinized using 0.25% trypsin, counted, and seeded on 24-well plates at a ratio of 1 × 10^5^ cells/well in the RPMI medium. The medium was changed every two days.

### 2.3. Preparation of PRF

For liquid PRF preparation, uncoated plastic PRF tubes (10 mL) were used. A total of 10 mL of blood per tube was collected from a peripheral vein using a vacuum blood collection butterfly and two different RCFs were used for PRF preparation: 600 rpm (44 g) and 8 min for low RCF and 2400 rpm (710 g) and 8 min for high RCF. The tubes with blood were placed immediately in the centrifuges (Intra Spin) after the blood withdrawal. To achieve a homogenous concentration of leukocytes, platelets, and growth factors, the liquid PRF was carefully pipetted from the PRF tubes to a sterile 15 mL plastic tube after centrifugation. The different-RCF PRF was then used for further experimentation and analysis as described in the following sections.

### 2.4. PRF Treatment of MG63 and HT1080

After HT1080 and MG63 were pre-seeded on 24-well plates (1 × 10^5^ cells/well) until reaching semi-confluence, high- and low-RCF PRF were prepared as described above. Before treating the cells with the different-RCF PRF, the medium in the wells was replaced with 1 mL of fresh RPMI medium. Depending on the experimental setting, 0.2 mL of the different-RCF liquid PRF was then quickly pipetted onto Thin Certs™ (Transwell filters) with a pore size of 0.4 µm, allowing a diffusion of growth factors and cytokines from the PRF to the tumor cells but hindering the passage of cells. Therefore, these Thin Certs™ were used for evaluating the indirect effect of different-RCF PRF on tumor cells by placing them on top of the seeded cells. After incubating the PRF in the transwells in the CO_2_ incubator for 10 min to allow clotting, 100 µL of the RMPI medium was added to the PRF clots in the upper compartment of the transwells. Each cell culture experiment had a control group which consisted of only cells in each well without PRF treatment. After 2 days of PRF treatment, supernatants were collected for ELISA, PRF clots were fixed in 4% paraformaldehyde for histology, and cells were analyzed for cell viability and lysed for RNA isolation and gene expression analyses.

### 2.5. Cell Viability Assay

To analyze the cell viability of the tumor cells in response to PRF treatment, the CellTiter^®^ 96 AQ One Solution Cell Proliferation Assay was used according to the manufacturer’s instruction. The cell viability of MG63 or HT1080 with or without treatment with the different-RCF PRF was evaluated using this method. After removing the RPMI medium from the wells, rinsing them with PBS, and aspirating the latter, 500 µL of fresh RPMI medium was added to each well before 100 µL/well of MTS (3-(4,5-dimethylthiazol-2-yl)-5-(3-carboxymethoxyphenyl)-2-(4-sulfophenyl)-2H-tetrazolium) was added and the plates were incubated in the CO_2_ incubator for 2 h. After incubation, 100 µL/well of the cell culture supernatant solution was transferred onto a 96-well plate and the optical density at a wavelength of 490 nm was measured using a microplate reader.

### 2.6. Immunofluorescence Staining

After removing the fixative (4% formaldehyde), the cells cultivated on the coverslips were washed three times with PBS, and 0.1% Triton-X in PBS was placed on the samples for 5 min for permeabilization of the cell membranes before washing again with PBS as described before. The primary antibody diluted in 1% BSA in PBS (anti-human SMA at a ratio of 1:2000) was used and incubated for one hour at room temperature before washing with PBS three times. A fluorescently labeled secondary antibody diluted in 1% BSA in PBS (Alexa^®^ 488 anti-mouse IgG at a ratio of 1:1000) was incubated for another hour in darkness at room temperature. After washing again with PBS, DAPI diluted in PBS (1:1000) was used for the counterstaining of the cell nuclei for 5 min before washing once with PBS. Finally, the coverslips were transferred to microscope slides and covered with Aquatex^®^ and microscope cover glasses. The examination of the stained cells was performed using a fluorescence microscope (Nikon Eclipse TS100, Düsseldorf, Germany).

### 2.7. Growth Factor and Cytokine Quantification with Enzyme-Linked Immunosorbent Assay (ELISA)

Supernatants were collected from different experiments, and the concentration of various growth factors and cytokines (transforming growth factor ß (TGFß), vascular endothelial growth factor (VEGF), and cellular communication network factors 1 and 3 (CCN1, CCN3)) were measured using ELISA DuoSets^®^ according to the manufacturers’ instructions in either duplicate or triplicate. For visualizing the concentration of the growth factors and cytokines, a streptavidin-HRP (horseradish-peroxidase) colorimetric reaction was used and the optical density was measured at wavelengths of 450 nm and 570 nm. The data measured at 570 nm were subtracted from the ones measured at 450 nm to correct the optical density.

### 2.8. Gene Expression Analyses

Isolating the RNA was carried out with the use of an RNeasy Micro Kit following the manufacturer’s protocol, and 1 µg of RNA per sample was transcribed into complementary DNA (cDNA) by following the manufacturer’s protocol using an Omniscript Reverse Transcription Kit. For 1 quantitative real-time polymerase chain reaction (qRT-PCR), 16 µL of SYBR^®^ Green Master Mix (2 µL of Primer 10× Qiagenmix, 10 µL of 2× SYBR^®^ Green Mix, and 4 µL of RNase free water) and 4 µL of cDNA (1 ng/µL), resulting in a total volume of 20 µL/well, were transferred to PCR plates, covered with foil, and briefly centrifuged. The primers that were used in this study include Glyceraldehyde 3-phosphate dehydrogenase (GAPDH), B-cell lymphoma 2 (BCL2), cyclin D1 and D2, p21, and p53. The cycler used in this study performs qRT-PCR using the following cycle program (40 cycles in total): denaturation at 94 °C for 15 s, annealing and elongation at 60 °C for 1 min, and dissociation at 94 °C for 15 s. Ribosomal protein 13A (RPL13A) was used as an endogenous standard and the relative gene expression of the samples was determined via the ΔΔCt method. Relative gene expression was compared by setting controls to 1 as a reference value.

### 2.9. Statistical Analysis

During this study, all experiments were performed with at least three different donors of PRF. The data presented are shown as mean values ± standard deviation and relative to the untreated control. Statistical significance was calculated with the one-way multifactorial variance analysis ANOVA test using graphed pad prism 9 (GraphPad Software Inc., San Diego, CA, USA). Statistical significance was assessed at * *p* < 0.05, ** *p* < 0.01, *** *p* < 0.001, and **** *p* < 0.0001 and documented in the figures.

## 3. Results

### 3.1. PRF-Mediated Effect on Cell Viability and Cell Morphology of Tumor Cells

Treating MG63 or HT1080 cells with low- and high-RCF PRF for 2 days resulted in a clear and significant reduction in the amount of viable tumor cells compared to untreated control cells (Figure 1A,B), as assessed by an MTS assay.

Although both RCF PRF applications lead to a significant reduction in viable cells, no significant difference could be documented between high- and low-RCF PRF with regard to the reduction in cell viability. Immunofluorescence staining of tumor cells using SMA, as a component of the cytoskeleton, corroborated the observed differences in cell quantity mediated by PRF treatment (Figure 2 and Figure 3). As demonstrated in Figure 2A–C, HT1080 cells without PRF treatment were densely organized in a confluent multilayer, fully covering the whole cover slips. Higher magnifications showed a rounded morphology of HT1080 in the control group (Figure 2C). Higher magnifications revealed a change in HT1080 morphology when treated with high-RCF PRF compared to the control (Figure 2F, arrow). Treatment of HT1080 with low-RCF PRF caused a more distinct reduction in the HT1080 cell amount, resulting in nearly empty coverslips in response to low-RCF PRF treatment (Figure 2G–I).

The effect of both RCF PRFs could be similarly observed on MG63 when compared to the control (Figure 3). MG63 cultivated without PRF had a spheroid shape with big cell nuclei and were organized in a dense cell network, filling the whole coverslip (Figure 3A–C). With the addition of high-RCF PRF to MG63, the cell numbers decreased, creating gaps between the cells (Figure 3D–F). After 2 days of treatment with high-RCF PRF, the size of the MG63 nuclei decreased in comparison to the MG63 control group and the cell morphology changed, resulting in cells exhibiting cuboid- or spindle-shaped morphology (Figure 3F, arrow). Treatment of MG63 with low-RCF PRF resulted in a significant reduction in observable MG63 cells (Figure 3G). Cell morphology tended towards the characteristic long or spindle shape of osteoblasts, with cells appearing smaller compared to those treated with high-RCF PRF (Figure 2H,I, asterisk). Both, high- and low-RCF PRF treatments led to a notable reduction in cell quantity.

### 3.2. Gene Expression Analyses of Cell Cycle- and Apoptosis-Associated Factors in MG63 and HT1080 in Response to PRF Treatment Compared to Untreated Controls

To understand the mechanisms that may lead to the PRF-mediated reductions in the cell viability of MG63 and HT1080, the expression of genes that are related to the cell cycle and apoptosis was determined in PRF-treated cells compared to controls without the application of PRF. The gene expression levels of cyclin D1, cyclin D2, GAPDH, BCL2, and p53 were assessed after 2 days of indirect high- and low-RCF PRF treatment. The results are demonstrated as relative gene expression compared to pure cell culture without treatment as the control (set to one). As part of glycolysis, GAPDH is usually associated with the breakdown of glucose for obtaining energy [15]. The determination of GAPDH’s relative gene expression showed a statistically significant up-regulation in MG63 treated with both RCF PRFs compared to pure cell culture (Figure 4B). The low-RCF PRF treatment of MG63 revealed no statistically significant different gene expression of GAPDH compared to high-RCF PRF with MG63. Contrarily, both RCF PRFs did not alter the expression of GAPDH in HT1080 significantly compared to the control (Figure 4A).

BCL2 protein is known to exert anti-apoptotic effects and its expression is usually up-regulated in tumor cells [16]. Quantitative RT-PCR revealed that PRF with MG63 led to a significant down-regulation of the relative gene expression of BCL2 as compared to the control (Figure 4D). MG63 cultivated with low-RCF PRF revealed significantly decreased expression of BCL2 in comparison to MG63 cultivated with high-RCF PRF (Figure 4D). On the other hand, HT1080 cultivated with high-RCF PRF led to a slightly higher BCL2 expression when compared to the control (Figure 4C).

HT1080 treated with low-RCF PRF revealed a higher expression of BCL2 than HT1080 treated with high-RCF PRF. Cyclins serve as checkpoints during the cell cycle and enable its termination in combination with cyclin-dependent kinases (CDKs); when present as a complex, constant high concentration of those cyclins can often be found in various tumor cells [17,18]. Expressions of cyclin D1 (Figure 4E) and cyclin D2 (Figure 4G) were distinctly down-regulated in HT1080 when cultivated with both RCF PRFs when compared to the controls without additional PRF, to an extent where the levels of cyclin D1 expression in HT1080 with PRF were leaning towards zero. PRF cultivated with MG63 also resulted in significantly down-regulated expression of cyclin D2 (Figure 4H) in comparison to the control.

Treatment with high-RCF PRF led to lower cyclin D2 expression in MG63 as compared to cyclin D2 expression when adding low-RCF PRF. The expression of cyclin D1 (Figure 4F) was significantly up-regulated in MG63 cultivated with PRF compared to the expression in the control. P53 is one of the key factors that decide upon the cell cycle route, and higher expressions are associated with a tumor-suppressive effect [19]. As depicted in Figure 4J, PRF treatment resulted in a significant up-regulated expression of p53 in MG63 as compared to p53 expression in the control. P53 was significantly more highly expressed in MG63 when treated with high-RCF PRF compared to the treatment of MG63 with low-RCF PRF.

Although the expression of p53 was found to be significantly up-regulated in HT1080 treated with low-RCF PRF compared to the control, the high-RCF PRF treatment of HT1080 resulted in a slightly, but not statistically significant, down-regulated expression of p53 when compared to the control (Figure 4I).

### 3.3. Determination of Growth Factors and Cytokines in Supernatants of MG63 and HT1080 in Response to Indirect PRF Application

The concentrations of the growth factors and cytokines TGF-ß1 and VEGF, as well as the tumor-related proteins CCN1 and CCN3, were evaluated in cell culture supernatants of MG63 and HT1080 in response to indirect PRF treatment for 2 days. Supernatants of pure cell cultures cultivated in a cell culture medium without PRF were used as controls. The aim of this experiment was the evaluation of indirect PRF-mediated changes in protein content release. As shown in Figure 5A,B, TFG-ß1 was present in all analyzed samples. Indirect application of low-RCF PRF to both tumor cell lines resulted in an increased release of TGF-ß1, calculated as significantly higher than the TGF-ß1 concentration within the supernatant of cells treated with high-RCF PRF and the controls. High-RCF PRF cultivated with MG63 or HT1080 resulted in a significantly decreased TGF-ß1 concentration in the supernatants compared to the cell control. The cultivation of low- and high-RCF PRF with MG63 led to significantly higher VEGF concentrations in the supernatants of the cells compared to the cell control (Figure 5D). HT1080 contained a generally higher concentration of VEGF in the supernatants compared to MG63 (Figure 5C).

Treating HT1080 with high-RCF PRF resulted in a decreased VEGF concentration within the supernatants compared to the HT1080 control group. Supernatants of HT1080 treated with low-RCF PRF resulted in lower VEGF concentrations in comparison to HT1080 cultivated with high-RCF PRF. As depicted in Figure 5F, MG63 treated with low-RCF or high-RCF PRF released a significantly higher concentration of CCN1 as compared to MG63 cultivated in a pure cell culture medium (control).

On the contrary, the treatment of HT1080 with PRF resulted in a non-statistically significant decrease in CCN1 concentrations in the supernatants when compared to HT1080 cultivated in a cell culture medium (Figure 5E). The addition of low-RCF PRF to HT1080 resulted in less CCN1 release in comparison to HT1080 with high-RCF PRF. Adding low-RCF or high-RCF PRF to MG63 did not change the concentrations of CCN3 in the supernatants compared to the control (Figure 5H). The HT1080 control groups contained significantly more CCN3 in cell culture supernatants than low-RCF and high-RCF PRF cultivated together with HT1080 (Figure 5G).

## 4. Discussion

Cancer remains the leading cause of death worldwide [20]. In the dental field, neoplasms are often found in the mucosa and can invade the underlying tissue, infiltrate the jaw bone, and finally spread throughout the body [21]. The treatment of cancer has been an extensive field of research for decades and has led to increased numbers of cured patients. However, especially when detected in a more progressive stage, treatment options remain insufficient for many patients and come along with high risks and side effects including hair loss, infection, pain, and mouth and throat problems.

In the context of potential tumor-suppressive impacts, the effect of injectable PRF was analyzed on in vitro monocultures of MG63 or HT1080 representing osteosarcoma (MG63) or fibrosarcoma (HT1080) cells [13,14]. Our findings could demonstrate decreased cell proliferation of MG63 and HT1080 when treated with PRF compared to cell cultures without PRF. The PRF-mediated effect was more distinct when the cells were treated with low-RCF PRF. According to this, gene and protein expression analyses revealed a consistent trend of up-regulation of tumor-suppressive genes and down-regulation of anti-apoptotic genes in both cell types following treatment with high- and, particularly, low-RCF PRF formulations.

In various clinical studies and in vitro studies, PRF showed its benefits in tissue regeneration and wound healing, since PRF seems to optimize the immune response as it provides a concentrated liquid of immune cells, growth factors, and cytokines [7,9]. The idea behind using PRF for the treatment of tumor cells is based on the principle that, usually, the organism can prevent the formation of neoplasms in the early stages [22].

The indirect PRF treatment of tumor cells resulted in decreased cell numbers of both tested cell types. In particular, the low-RCF PRF treatment caused a significant decrease in the number of viable cells in both cell types that could also be confirmed visually using immunofluorescence staining. The main components of PRF are platelets, leukocytes (including neutrophils), stem cells, fibrin, and bioactive molecules [7]. While neutrophils are the first line of defense against pathogens, they are also able to infiltrate tumors, kill tumor cells, and mediate tumor cytotoxicity [23]. Apart from that, neutrophils communicate closely with monocytes and macrophages. In wounds, macrophages phagocytose remnants of destroyed cells and aid in ‘cleaning up’ the inflammation after neutrophils secrete a variety of proteases degrading extracellular components, and might also be a potent contributor to the paracrine-mediated effect of PRF on tumor cells. Moreover, PRF contains T- and B-lymphocytes, which are usually more prominent towards the last phases of inflammation and wound healing, responsible for killing tumor cells, mainly by binding to death receptors and activating the extrinsic pathway of apoptosis [24].

The largest component of PRF consists of fibrin and platelets, fragments derived from megakaryocytes that are essential for hemostasis, inflammation, and immunity [25]. Platelets release huge amounts of bioactive molecules upon activation and degranulation. Further, platelets are able to induce the death of infected cells via complex mechanisms [26,27]. Since platelets are cell fragments with an average size of 2–4 µm in diameter, they can migrate through the pores of the membranes used in our indirect experiments. As a consequence, they might also be responsible for the toxic effect of PRF on MG63 or HT1080 in the indirect experiments.

A broad and complex network of mechanisms contributes to the development of cancer, and usually, the human organism can interfere at different stages to eliminate (pre-) malignant cells [28]. The inhibition of cell cycle progression is one such protective mechanism and is mediated by certain tumor suppressors and growth factors [29]. The treatment of tumor cells with low-RCF PRF led to significantly higher TGF-ß1 concentrations compared to the cell control of both cell types. TGF-ß1 holds a significant influence in cell cycle regulation, leading to growth inhibition and the induction of apoptosis, particularly in the presence of VEGF [30]. Generally, VEGF is up-regulated in cancer, and some treatment approaches include the inhibition of VEGF receptors [31]. Indirect treatment with low-RCF PRF led to a significantly higher VEGF release in MG63 compared to the untreated cell control. Thus, the combination with other growth factors or cytokines in this dynamic system might contribute to an inhibiting impact on tumor cells.

Senescence and apoptosis might be enhanced in MG63 since up-regulated CNN1 expression is connected to the activation of tumor-suppressive genes like p53 and Rb protein [32]. Possibly, the interactions between CCN1 and other stimuli might change the balance between the pro-apoptotic effect and growth-promoting effect towards the former in MG63 treated with PRF. CCN3 seemed to be significantly more highly expressed in MG63 and HT1080 compared to PRF cultivated without cells. It remains an open question as to whether small changes in CNN3 expression are sufficient to contribute to cell death or whether they do not play a major role in that regard.

Interestingly, our findings revealed an up-regulation of GAPDH expression in both MG63 and HT1080 treated with PRF compared to the control groups without PRF. Since growth factors can act as mitotic stimuli for cells, exceedingly high mitotic signals might trigger cell death even in malignant cells [33]. Nonetheless, studies indicate that GAPDH possesses broader functionalities beyond its role in glycolysis. It can translocate into the nucleus and might contribute to chromosome stability, but in response to oxidative stress, GAPDH may help to initiate apoptosis [34]. Furthermore, there is evidence that GAPDH expression is up-regulated through increased levels of p53, indicating the correlation of apoptosis and higher amounts of GAPDH [35]. In accordance with our suggestion that PRF might induce apoptosis in tumor cells, the relative expression of p53 was significantly up-regulated in MG63 treated with PRF compared to MG63 without PRF. The so-called ‘guardian of the genome’, p53, as a transcription factor, is essential for physiological cell proliferation, and thus mutates in the majority of human carcinomas [19]. In particular, low-RCF PRF application to MG63 and HT1080 led to 2-fold up-regulated p53 expression compared to the control groups. The higher concentrations of TGF-ß1 observed in HT1080 treated indirectly with low-RCF PRF compared to the control group may contribute to the up-regulated expression of p53 [36]. However, not only up-regulated expressions of tumor-suppressive genes are necessary for the induction of apoptosis, but also, the down-regulated expression of (proto-) oncogenes drive the cell towards death. Bcl2, as an important proto-oncogene, was significantly down-regulated in MG63 cultivated with PRF compared to the controls. According to several studies, p53 plays a major role in the regulation of Bcl2 expression [37].

The decision to enter the cell cycle in the early G phase is highly dependent on the availability of D-type cyclins activating Cdk 4 and Cdk 6 [18]. The relative gene expression of cyclin D1 was extremely down-regulated in HT1080 treated with both RCF PRFs compared to the control cultivated without PRF. Similarly, the expression of cyclin D2 was also significantly down-regulated in HT1080 and MG63 treated with PRF as compared to the control groups. Since the down-regulation of cyclins causes cell cycle progression to stop, the evaluation of decreased proliferation in response to PRF seems to fit with those results. Since p53 was highly expressed in MG63 and HT1080 treated with low-RCF PRF, the measured results of down-regulated cyclin D1 and D2 expression confirm this suggestion from previous studies [38]. Still, PRF the treatment of MG63 led to significantly up-regulated expression of cyclin D1. Studies have shown that while moderate D-type cyclin up-regulation leads to Cdk activation and contributes to cell cycle progression, overexpression of those cyclins results in halting the cell at the G1/S checkpoint. It might be possible that MG63 is more sensitive to the changes regarding growth factor or cytokine concentrations, while HT1080 reacts through other pathways initiated by PRF treatment. Apparently, HT1080 and MG63 seem to react differently to PRF treatment regarding changes in the expression of mitotic stimuli, i.e., growth factors or cytokines. Furthermore, even similar external stimuli can cause completely different results regarding the outcomes of gene expression of p53, cyclins, or Bcl2.

Although the molecular processes vary between both tumor cell types, it is remarkable that the final observed outcome still leads to decreased cell proliferation, increased cell death, and morphological changes. However, our in vitro observations showed that PRF, especially low-RCF PRF, can destroy tumor cells and might therefore be an additional treatment option for localized tumors in the future. Nevertheless, the immunomodulating properties of PRF should be examined very critically, since PRF, especially when generated from tumor patients, might also favor tumor promotion or metastasis. However, the observations in this study are highly promising and should pave the way for future research on PRF as a patient-specific immunomodulatory cell culture system. Therefore, further investigations of possible molecular changes and extensions of this study to include more types of tumor cells or primary tumor cells are necessary for potentially using PRF as an co-adjuvant therapy, particularly for treating local tumors with a manageable size in the oral cavity or the skin in the future.

## Figures and Tables

**Figure 1 bioengineering-11-00303-f001:**
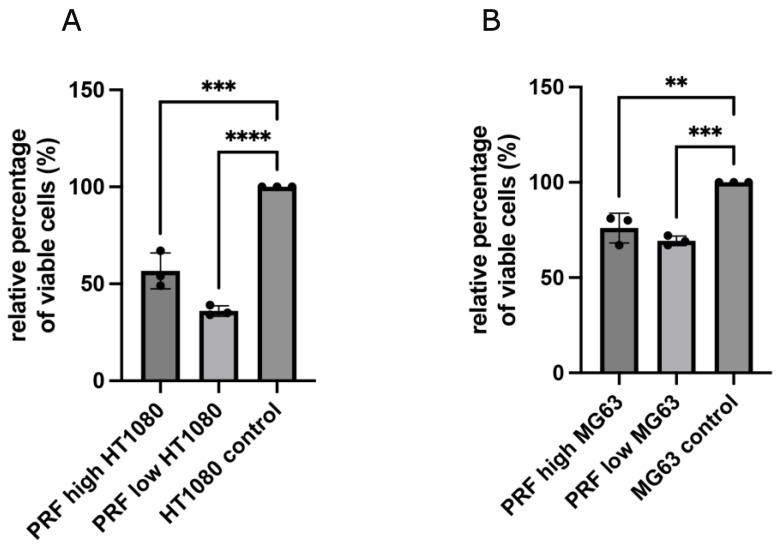
Results of CellTiter^®^ 96 AQ One Solution Cell Proliferation Assay. Comparing the effect of low- and high-RCF PRF on cell viability of HT1080 (**A**) and MG63 (**B**) compared to pure cell culture using transwells. Results are presented as a relative percentage of viable cells. Cells cultivated without PRF (control) were set to 100%. Statistical significance was calculated with the one-way multifactorial variance analysis ANOVA test. Statistical significance was assessed when ** *p* < 0.01, *** *p* < 0.001, and **** *p* < 0.0001.

**Figure 2 bioengineering-11-00303-f002:**
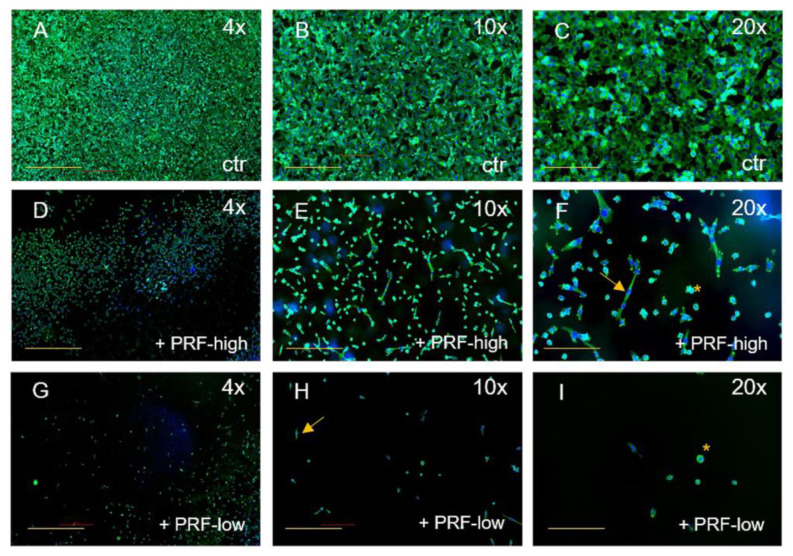
Immunofluorescence staining of HT1080 for SMA (green) and DAPI (blue). Comparison of HT1080 without PRF (**A**–**C**), treated with high-RCF PRF (**D**–**F**), and treated with low-RCF PRF (**G**–**I**) after 2 days of cultivation. Arrows: spindle-shaped cell morphology. Scale bars: (**A**,**D**,**G**) = 500 µm, (**B**,**E**,**H**) = 200 µm, (**C**,**F**,**G**) = 100 µm.

**Figure 3 bioengineering-11-00303-f003:**
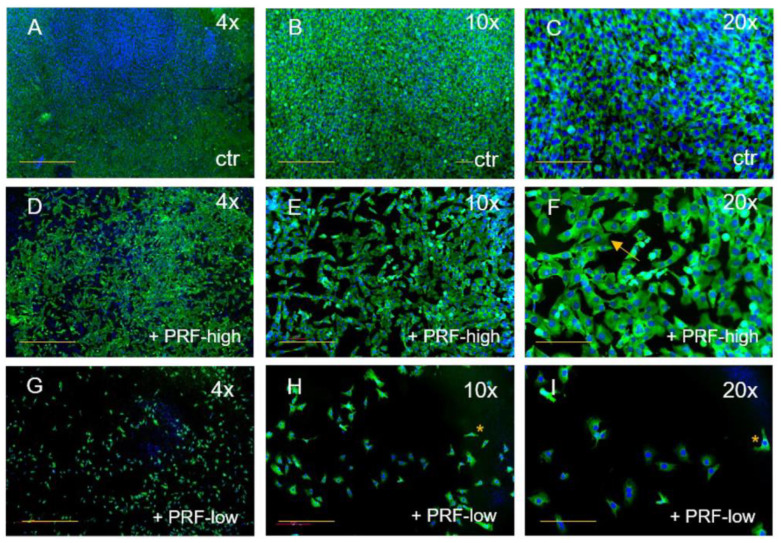
Immunofluorescence staining of MG63 for SMA (green) and DAPI (blue). Comparison of MG63 without PRF (**A**–**C**), treated with high-RCF PRF (**D**–**F**), and treated with low-RCF PRF (**G**–**I**) after 2 days of cultivation. Arrows: spindle-shaped cell morphology. Scale bars: (**A**,**D**,**G**) = 500 µm, (**B**,**E**,**H**) = 200 µm, (**C**,**F**,**G**) = 100 µm. * more detail in Figure 2.

**Figure 4 bioengineering-11-00303-f004:**
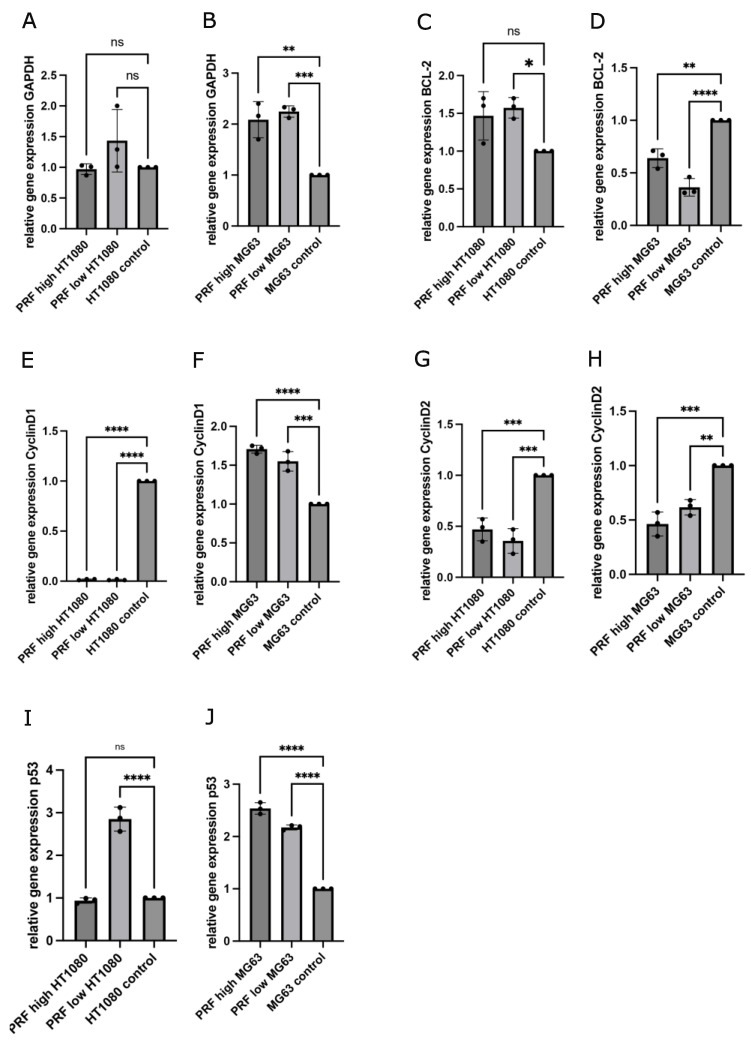
Determination of relative gene expression of GAPDH (**A**,**B**), BCL-2 (**C**,**D**), cyclin D1 (**E**,**F**), cyclin D2 (**G**,**H**), and p53 (**I**,**J**) using quantitative real-time-PCR comparing changes in gene expression of MG63 and HT1080 cultivated with low- or high-RCF PRF in contrast to cells without PRF (control = 1). Statistical significance was calculated with the one-way multifactorial variance analysis ANOVA test. Statistical significance was assessed when * *p* < 0.05, ** *p* < 0.01, *** *p* < 0.001, and **** *p* < 0.0001, ns = not significant.

**Figure 5 bioengineering-11-00303-f005:**
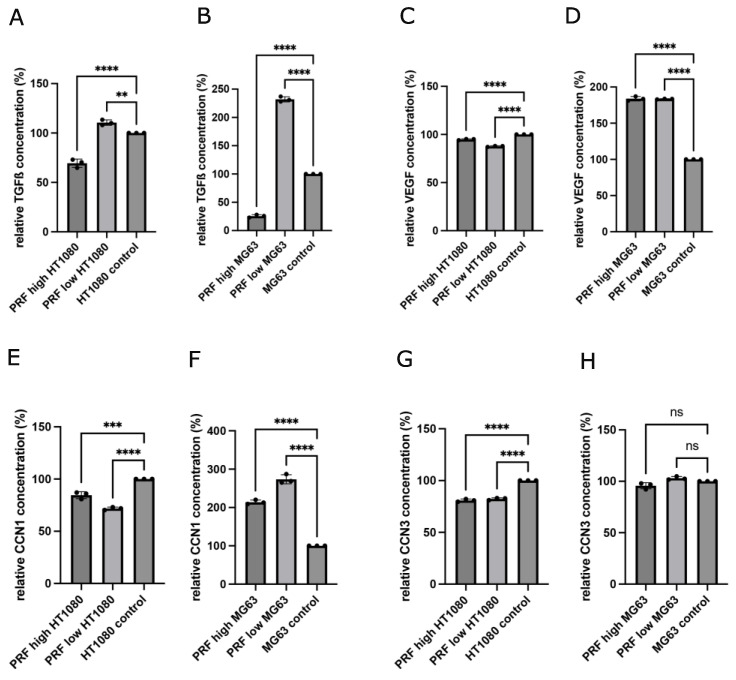
Quantification of growth factor and cytokine concentrations of TGF-ß1 (**A**,**B**), VEGF (**C**,**D**), CCN1 (**E**,**F**), and CCN3 (**G**,**H**) within supernatants released from indirectly treated HT1080 and MG63 with low- and high-RCF PRF (after 2d of cultivation). Results are calculated as relative concentrations compared to control (set to 100%). Statistical significance was calculated with the one-way multifactorial variance analysis ANOVA test. Statistical significance was assessed when ** *p* < 0.01, *** *p* < 0.001, and **** *p* < 0.0001, ns = not significant.

## Data Availability

The data that support the findings of this study are available upon request from the corresponding author (E.D.).

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
