# Peer review of "Immunomodulation of Cancer Cells Using Autologous Blood Concentrates as a Patient-Specific Cell Culture System: A Comparative Study on Osteosarcoma and Fibrosarcoma Cell Lines"

_bioengineering, 2024, doi:10.3390/bioengineering11040303_

Round 1

Reviewer 1 Report

Comments and Suggestions for Authors

Abstract

In the abstract, acronyms are used without prior definition, so the abstract is not intelligible to anyone not working in this field (or in this research group). It is therefore useless to the general reader. Please define these materials before referring to them by abbreviations. If this proves difficult because it will exceed the abstract limit for words, simply refer to them as “concentrates”, for the reader will surely be able to judge whether the article is of interest.

Introduction

Introduction. “PRF as an autologous concentrate of the blood’s immune properties, i.e. leukocytes, growth factors, cytokines, platelets and fibrin, might therefore be an additional treatment option for localized tumors. Especially, for visible carcinomas with a manageable size like in the oral cavity or the skin, PRF might be easily applied without side effects since ‘the
drug’ is the body’s immune components in a concentrated form. PRF, as a booster for wound healing and regeneration, may be a promising additional treatment option for cancer patients to enhance the body's immune response and accelerate healing mechanisms [12].” This reasoning is one-sided, and should be expanded to include the possibility that any such biomaterial, such as this complex mix, may have tumor-promoting activities. An extensive literature exists on immune mechanisms favoring tumor evasion and curbing effective antitumor immunity in vivo. It is fine if this treatment works, but it might just as well be ineffective, of even deleterious.

Methods

“In addition, two further experimental groups, pure low- and high-RCF PRF, also served as controls, depending on the experiment.” These two experimental groups are not described in sufficient detail in the Methods section to help the reader assess their usefulness. If this depends on the experiment, as stated by the authors, if would perhaps be easier to mention them in the Results section, when the data in all groups are mentioned.

The section on statistical methods mentions “relative mean values ± standard deviation”; is that different from “ mean values ± standard deviation”? If so, please clarify. If not, please avoid the “relative” qualification, so as not to confuse the reader.

Results

Figure 1 caption mentions “relative percentage of viable cells.” As in the above query for the Methods section, the term “relative” is of unclear interpretation. By looking at the figure, I judged it to present data as percentage of viable cells in different groups, which is fine. By adding “relative” is the interpretation somehow changed? Please clarify.

“While the impact of low-RCF PRF appeared more pronounced in reducing cell proliferation compared to high-RCF PRF, this disparity did not reach statistical significance.” This phrasing suggests that the authors use statistics but do not believe statistics. The lack of statistical significance means that (in their present state) their data are not sufficient to reject the null hypothesis. With more data, or with different data, one might perhaps reject the null hypothesis and present a significant P value. So, being sober on the description of their opinion will be helpful to the reader. No, they could not demonstrate a significant difference in this specific analysis. Yes, they cannot exclude that future experiments may unveil significant differences in a future analysis. Would they be justified in sticking to their gut feeling? Yes, as long as they phrase it as a hope rather than as an underrated “fact”. With the necessary changes, the above criticism would just as well apply to the following statement, found elsewhere in the manuscript: “Low-RCF PRF treatment of MG63 revealed a slightly higher, but statistically not significant, expression of GAPDH compared to high-RCF PRF with MG63”. A further example of what seems to be an established way of thinking about undesired results is found further down: “While additional low-RCF PRF resulted in a slightly higher expression of GAPDH in HT1080 in comparison to high-RCF PRF with HT1080, the difference was not significant.” That is exactly what random fluctuation usually is - a “slight” difference that turns out to be “nonsignificant”, i. e., not entailing rejection of the null hypothesis.

On the basis of these (out many) examples, I would strongly recommend that the authors decide on whether they want to describe their data on the grounds of statistically significant results. If so, they cannot suggest that the results of these statistical tests are regrettable. That is not to suggest that a biological study cannot be described without statistical analysis, but in this case it loses the credibility that is commonly attributed to the correct use of statistics.

Discussion.

Considering the contents of the Results section, I found the Discussion section, which takes 3 and one-half pages in the printed manuscript, somewhat excessive. It is unclear to me whether it could benefit from a focus on the experimental findings and a lesser concern with the relationship of their findings to a vast literature. A more focused discussion, in my view, is unlikely to occupy 3 1/2 pages, but is likely to help the reader appreciate the original contribution of the reported experiments.

Author Response

Dear Reviewer,

we would like to thank you for carefully reading the manuscript and for your response. Please find included the resubmission of the manuscript entitled:

“Immunomodulation of cancer cells using autologous blood concentrates as patient-specific cell culture system: a comparative study on osteosarcoma and fibrosarcoma cell lines”

by Eva Dohle, Kamelia Parkhoo, Francesco Bennardo, Lena Schmeinck, Robert Sader and Shahram Ghanaati,

to be considered for publication as original research paper in the Journal Bioengineering. We have revised the manuscript according to your suggestions. The changes are addressed in this letter. We would like to thank you for all your effort with the manuscript.

Yours sincerely,

Eva Dohle

General information:

The individual answers to the reviewers’ suggestions are addressed in this letter. All changes in the revised manuscript have been highlighted in yellow colour.

Abstract 

In the abstract, acronyms are used without prior definition, so the abstract is not intelligible to anyone not working in this field (or in this research group). It is therefore useless to the general reader. Please define these materials before referring to them by abbreviations. If this proves difficult because it will exceed the abstract limit for words, simply refer to them as “concentrates”, for the reader will surely be able to judge whether the article is of interest.

According to this suggestion, we defined ‘PRF’ and ‘RCF’ in the abstract before referring to them as abbrevations in the abstract.

Introduction. “PRF as an autologous concentrate of the blood’s immune properties, i.e. leukocytes, growth factors, cytokines, platelets and fibrin, might therefore be an additional treatment option for localized tumors. Especially, for visible carcinomas with a manageable size like in the oral cavity or the skin, PRF might be easily applied without side effects since ‘the
drug’ is the body’s immune components in a concentrated form. PRF, as a booster for wound healing and regeneration, may be a promising additional treatment option for cancer patients to enhance the body's immune response and accelerate healing mechanisms [12].” This reasoning is one-sided, and should be expanded to include the possibility that any such biomaterial, such as this complex mix, may have tumor-promoting activities. An extensive literature exists on immune mechanisms favoring tumor evasion and curbing effective antitumor immunity in vivo. It is fine if this treatment works, but it might just as well be ineffective, of even deleterious.

Accordingly, we expanded the introduction (as well as the discussion part) with regard to this comment.

Methods

“In addition, two further experimental groups, pure low- and high-RCF PRF, also served as controls, depending on the experiment.” These two experimental groups are not described in sufficient detail in the Methods section to help the reader assess their usefulness. If this depends on the experiment, as stated by the authors, if would perhaps be easier to mention them in the Results section, when the data in all groups are mentioned.

We deleted this information according to this suggestion.

The section on statistical methods mentions “relative mean values ± standard deviation”; is that different from “mean values ± standard deviation”? If so, please clarify. If not, please avoid the “relative” qualification, so as not to confuse the reader.

Thank you for this suggestion. The reason for describing the statistic/diagrams as relative mean values is based on the calculation of all results (MTS, gene expression, protein concentration) by referring to the control (untreated cells). That means that we are not showing the ‘absolute’ concentrations, values or gene expression but relative to the control cells (that were set to 100% or to 1).

Results

Figure 1 caption mentions “relative percentage of viable cells.” As in the above query for the Methods section, the term “relative” is of unclear interpretation. By looking at the figure, I judged it to present data as percentage of viable cells in different groups, which is fine. By adding “relative” is the interpretation somehow changed? Please clarify.

Thank you for this suggestion. The reason for describing the statistic/diagrams as relative mean values is based on the calculation of all results (MTS, gene expression, protein concentration) by referring to the control (untreated cells). That means that we are not showing the ‘absolute’ concentrations, values or gene expression but relative to the control cells (that were set to 100% or to 1).

“While the impact of low-RCF PRF appeared more pronounced in reducing cell proliferation compared to high-RCF PRF, this disparity did not reach statistical significance.” This phrasing suggests that the authors use statistics but do not believe statistics. The lack of statistical significance means that (in their present state) their data are not sufficient to reject the null hypothesis. With more data, or with different data, one might perhaps reject the null hypothesis and present a significant P value. So, being sober on the description of their opinion will be helpful to the reader. No, they could not demonstrate a significant difference in this specific analysis. Yes, they cannot exclude that future experiments may unveil significant differences in a future analysis. Would they be justified in sticking to their gut feeling? Yes, as long as they phrase it as a hope rather than as an underrated “fact”. With the necessary changes, the above criticism would just as well apply to the following statement, found elsewhere in the manuscript: “Low-RCF PRF treatment of MG63 revealed a slightly higher, but statistically not significant, expression of GAPDH compared to high-RCF PRF with MG63”. A further example of what seems to be an established way of thinking about undesired results is found further down: “While additional low-RCF PRF resulted in a slightly higher expression of GAPDH in HT1080 in comparison to high-RCF PRF with HT1080, the difference was not significant.” That is exactly what random fluctuation usually is - a “slight” difference that turns out to be “nonsignificant”, i. e., not entailing rejection of the null hypothesis..

On the basis of these (out many) examples, I would strongly recommend that the authors decide on whether they want to describe their data on the grounds of statistically significant results. If so, they cannot suggest that the results of these statistical tests are regrettable. That is not to suggest that a biological study cannot be described without statistical analysis, but in this case it loses the credibility that is commonly attributed to the correct use of statistics.

We adjusted the results part according to this suggestion and deleted unnecessary phrasing

Discussion. 

Considering the contents of the Results section, I found the Discussion section, which takes 3 and one-half pages in the printed manuscript, somewhat excessive. It is unclear to me whether it could benefit from a focus on the experimental findings and a lesser concern with the relationship of their findings to a vast literature. A more focused discussion, in my view, is unlikely to occupy 3 1/2 pages, but is likely to help the reader appreciate the original contribution of the reported experiments.

According to this suggestion, we shortened the discussion part to less than 3 pages. Since we analysed the effect of 2 different PRFs on 2 different tumor cell lines with regard to a lot of factors etc, it would be very difficult to further shorten the discussion. For a better understanding and not to lose the focus, we added some sentences to clarify the implications of the findings.

Reviewer 2 Report

Comments and Suggestions for Authors

1. Introduction:

  1. Consider providing a brief definition or explanation of PRF (Platelet-Rich Fibrin) upon first mention, for readers who may not be familiar with the term.
  2. It would be beneficial to mention the full forms of abbreviations like TGF-ß1 and VEGF the first time they are used in the introduction to aid comprehension for readers who may not be familiar with these terms.
  3. In the first sentence, consider revising to: "In the medical and dental fields, continuous development of regenerative concepts is essential to ensure tissue regeneration through effective wound healing..."
  4. Ensure consistency in punctuation throughout the text. For example, use consistent spacing after periods and commas.
  5. Consider revising the first sentence for clarity: "In the medical and dental field, continuous development of regenerative concepts is essential to ensure tissue regeneration after trauma or tumor resection."
  6. Specify what "RCF" stands for upon first mention.
  7. Consider splitting the last sentence for better readability.

2. Results:

  1. In the first paragraph, clarify what MTS assay is upon first mention.
  2. The results section provides a comprehensive analysis of the effects of PRF on cell viability, morphology, and gene expression in tumor cell lines MG63 and HT1080. However, it might be helpful to briefly summarize the key findings at the beginning of this section to provide an overview before delving into the details.
  3. In Figures 2 and 3, labeling the axes of the graphs with the corresponding units (e.g., time in days, cell count, etc.) would enhance clarity.
  4. It would be helpful to provide statistical analysis methods (e.g., ANOVA followed by post-hoc tests) used to compare different treatments and controls in the results section, particularly when stating the significance of the results.
  5. "Treating MG63 or HT1080 cells with low and high-RCF PRF" - Consider revising to: "Treating MG63 or HT1080 cells with low- and high-RCF PRF" for clarity.
  6. Consider breaking down the large paragraph discussing cell morphology observations for better readability.
  7. In the gene expression analysis section, clarify the significance of the results for each gene studied.

3. Discussion:

  1. The discussion provides a thorough interpretation of the results and relates them to the existing literature. However, consider expanding on the potential clinical implications of the findings. How might these results impact the development of PRF-based therapies for cancer treatment?
  2. Provide more context on the limitations of the study, such as the use of cell lines instead of primary tumor cells or the need for further in vivo validation of the observed effects.
  3. "However, at least our in vitro observations showed that PRF, especially low-RCF PRF, can destroy cells and in tumor-like tissue and might therefore be an additional treatment option for localized tumors in the future." - This sentence is a bit convoluted. Consider revising for clarity, perhaps breaking it into two sentences.
  4. The discussion is comprehensive, but it could benefit from a more concise presentation of key findings.
  5. Some paragraphs are quite long and could be split for easier digestion of information.
  6. Consider providing a summary of the main findings before delving into the complexities of the results.
  7. Clarify the implications of the findings for future research and clinical applications.
Comments on the Quality of English Language

Moderate editing of the English language required

Author Response

Dear Reviewer,

we would like to thank you for carefully reading the manuscript and for your response. Please find included the resubmission of the manuscript entitled:

“Immunomodulation of cancer cells using autologous blood concentrates as patient-specific cell culture system: a comparative study on osteosarcoma and fibrosarcoma cell lines”

by Eva Dohle, Kamelia Parkhoo, Francesco Bennardo, Lena Schmeinck, Robert Sader and Shahram Ghanaati,

to be considered for publication as original research paper in the Journal Bioengineering. We have revised the manuscript according to your suggestions. The changes are addressed in this letter. We would like to thank you for all your effort with the manuscript.

Yours sincerely,

Eva Dohle

General information:

The individual answers to the reviewers’ suggestions are addressed in this letter. All changes in the revised manuscript have been highlighted in yellow colour.

  1. Introduction:

  1. Consider providing a brief definition or explanation of PRF (Platelet-Rich Fibrin) upon first mention, for readers who may not be familiar with the term.

We provided a brief definition of “PRF” when first used in the manuscript, accordingly.

  1. It would be beneficial to mention the full forms of abbreviations like TGF-ß1 and VEGF the first time they are used in the introduction to aid comprehension for readers who may not be familiar with these terms.

The full forms of TGF, VEGF can already be found in the introduction part in line 48-50. This is the first time they are used in the manuscript.

  1. In the first sentence, consider revising to: "In the medical and dental fields, continuous development of regenerative concepts is essential to ensure tissue regeneration through effective wound healing..."

The sentence has been changed, accordingly.

  1. Ensure consistency in punctuation throughout the text. For example, use consistent spacing after periods and commas.

The manuscript has been checked and revised again.

  1. Consider revising the first sentence for clarity: "In the medical and dental field, continuous development of regenerative concepts is essential to ensure tissue regeneration after trauma or tumor resection."

The sentence has been changed, accordingly.

  1. Specify what "RCF" stands for upon first mention.

The abbreviation ‘RCF’ has been defined when first used in the manuscript.

  1. Consider splitting the last sentence for better readability.

According to this suggestion, the sentence has been spitted.

  1. Results:

  1. In the first paragraph, clarify what MTS assay is upon first mention.

According to this suggestion, “MTS” has been clarified and written as (3-(4,5-dimethylthiazol-2-yl)-5-(3-carboxymethoxyphenyl)-2-(4-sulfophenyl)-2H-tetrazolium) when first used.

  1. The results section provides a comprehensive analysis of the effects of PRF on cell viability, morphology, and gene expression in tumor cell lines MG63 and HT1080. However, it might be helpful to briefly summarize the key findings at the beginning of this section to provide an overview before delving into the details.

The authors decided not to summarize again the results in general at the beginning of the results part. The results of each experiment are summarized as well as detailed described in each of the result paragraphs.

  1. In Figures 2 and 3, labeling the axes of the graphs with the corresponding units (e.g., time in days, cell count, etc.) would enhance clarity.

According to this suggestion, we added information regarding the time of cultivation to the appropriate figure legends.

  1. It would be helpful to provide statistical analysis methods (e.g., ANOVA followed by post-hoc tests) used to compare different treatments and controls in the results section, particularly when stating the significance of the results.

We provided information on the statistical analysis methods in the appropriate figure legends, accordingly.

  1. "Treating MG63 or HT1080 cells with low and high-RCF PRF" - Consider revising to: "Treating MG63 or HT1080 cells with low- and high-RCF PRF" for clarity.

This has been corrected.

  1. Consider breaking down the large paragraph discussing cell morphology observations for better readability.

According to this suggestion, the paragraph has been shortened.

  1. In the gene expression analysis section, clarify the significance of the results for each gene studied.

The information regarding significance of the gene expression results has been added to the figure legend.

  1. Discussion:

  1. The discussion provides a thorough interpretation of the results and relates them to the existing literature. However, consider expanding on the potential clinical implications of the findings. How might these results impact the development of PRF-based therapies for cancer treatment?

Information on the potential clinical implications of the findings has been added to the discussion part, accordingly.

  1. Provide more context on the limitations of the study, such as the use of cell lines instead of primary tumor cells or the need for further in vivo validation of the observed effects.

Furthermore, a paragraph concerning the limitation and planned further research has been added to the discussion part.

  1. "However, at least our in vitro observations showed that PRF, especially low-RCF PRF, can destroy cells and in tumor-like tissue and might therefore be an additional treatment option for localized tumors in the future." - This sentence is a bit convoluted. Consider revising for clarity, perhaps breaking it into two sentences.

According to this suggestion, the sentences has been revised.

  1. The discussion is comprehensive, but it could benefit from a more concise presentation of key findings.

The key findings have been summarized in the second paragraph of the discussion part.

  1. Some paragraphs are quite long and could be split for easier digestion of information.

In general, the discussion part has been shortened and some paragraphs have been splitted, accordingly.

  1. Consider providing a summary of the main findings before delving into the complexities of the results.

The key findings have been summarized in the second paragraph of the discussion part.

  1. Clarify the implications of the findings for future research and clinical applications.

Information on the potential clinical implications of the findings has been added to the discussion part, accordingly.
